# Kinetic and Equilibrium Isotherm Studies for the Removal of Tetracycline from Aqueous Solution Using Engineered Sand Modified with Calcium Ferric Oxides

Osamah Al-Hashimi [1,2,*], Khalid Hashim [2], Edward Loffill [2], Ismini Nakouti [3], Ayad A. H. Faisal [4] and Tina Marolt Čebašek [2]

1   Babylon Water Directorate, Babylon 51001, Iraq
2   School of Civil Engineering and Built Environment, Liverpool John Moores University, Liverpool L3 3AF, UK
3   Built Environment and Sustainable Technology Research Institute, Liverpool John Moores University, Byrom Street, Liverpool L3 3AF, UK
4   Department of Environmental Engineering, College of Engineering, University of Baghdad, Baghdad 10001, Iraq
*   Correspondence: o.a.alhashimi@2020.ljmu.ac.uk; Tel.: +44-752-2668404

**Abstract:** The novel aspect of this research is the fabrication, characterisation, and application of an engineered adsorbent made from quartz sand coated with calcium ferric oxides (QS/CFO) derived from the wastepaper sludge ash (WPSA) for the removal of tetracycline (TC) from synthetic water. Initially, the new adsorbent was fabricated using a Ca/Fe molar ratio, sand/FeCl$_3$ ratio, pH (of synthesising environment), ethylene glycol dose, and temperature of 1:0.75, 1:1, 12, 6 mL/100 mL, and 95 °C, respectively. Then, the new adsorbent was applied to treat water having 50 mg/L of TC in batch experiments, taking into account the effects of the contact time (0–180 min), pH of water (2–12), the dose of adsorbent (0.05–0.5 g), and agitation speed (0–250 rpm). The results obtained proved the engineered adsorbent can remove as much as 90% of the TC (adsorption capacity of 21.96 mg/g) within 180 min at an initial pH, adsorbent dosage, and agitation speed of 7, 0.3 g per 50 mL, and 200 rpm, respectively. It was also found that the pseudo-second-order model describes the kinetic measurements better than the pseudo-first-order model, which indicates that the TC molecules have been bonded with the prepared sorbent through chemical forces. Furthermore, the intra-particle diffusion model results demonstrated that the diffusion mechanism plays a significant role in TC adsorption; however, it was not the predominant one. Finally, the outcomes of the characterisation analysis proved that the newly formed layer on the quartz sand substantially contributed to the removal of the TC from the contaminated water.

**Keywords:** tetracycline; kinetics of adsorption; batch tests; calcium ferric oxid; coated sand; wastepaper sludge ash

## 1. Introduction

Tetracycline (TC) is a broad-spectrum antibiotic which consists of 59.4540% of carbon (C), 5.4430% hydrogen (H), 6.3032% nitrogen (N), and 28.800% of oxygen (O) [1], with amphoteric, phenolic, and alcoholic properties [2]. Owing to the presence of a dimethylam-monium group, a phenolic diketone moiety, and a tricarbonyl system, TC exists primarily as cationic (TCH$_3^+$) at pH less than 3.3, zwitterionic (TCH$_2^\pm$) at pH between 3.3 and 7.7, and anionic (TCH$^+$) and (TCH$_2^-$) at pH greater than 7.7 [3,4]. TC is massively produced due to its extensive use in treating a wide range of bacterial infections [5]. Additionally, it is widely used for veterinary and agricultural purposes [6,7]. Unfortunately, the excessive use of antibiotics threatens the environment and public health. In fact, Germany, China, the EU, and Switzerland are among the countries with relatively high TC usage. For instance, in 2013, China used 6950 tonnes of TC-based antibiotics [8]. The consumption of TC-based

antibiotics in Germany in 2016 reached 542 tonnes. In 2017, the United Kingdom used 104.9 tonnes of TC-based antibiotics for animal treatment and 48.2 tonnes for human treatment [9]. This vast global usage of TC-based antibiotics is the main reason for the presence of TC in surface waters [10].

The presence of TC in water is a problematic issue for the water industry due to the complex composition of the TC, which prohibits the efficient removal of TC in conventional wastewater treatment facilities [11]. As with any other organic chemical compound, TC-based antibiotics undergo various physical, chemical, and biological processes in the aquatic environment that lead to the partial or complete degradation of the TC-based antibiotics. The level and pace of the degradation process depend on the physicochemical characteristics of the TC-based antibiotics, the characteristics of the bed soil, and the ambient conditions, such as the temperature and presence of other chemicals [12]. During the degradation journey in the aquatic environment, the TC-based antibiotics may migrate from one phase to another (from water to soil and vice versa) due to the adsorption and desorption, leaching, drainage, absorption by crops, dispersion, and volatilisation. Furthermore, the structure of TC-based antibiotics may alter to another structure during this journey through transformation processes, such as hydrolysis, photocatalytic degradation, bioremediation, oxidisation, and reduction. TC may have a partitioning, distribution, and adsorption coefficient of 15,278 L/kg [10] and, in some other forms, may reach 312,447 L/kg [13]. Bao et al. [14] confirmed tetracycline may have a partitioning, distribution, and adsorption coefficient of 15,278 L/kg, and may reach 312,447 L/kg in some forms [13]. This demonstrates that TCs are very durable and non-degradable substances because any compound with a sorption coefficient of 4000 L/kg is considered very resistant and non-degradable [15].

Although many methods are used to remove TC-based antibiotics from wastewater and water, such as biodegradation, they do not enjoy high removal efficiency. Currently, adsorption is the predominant method for removing TC-based antibiotics from water and wastewater [16,17]. Several materials have been used as adsorbents to remove TC from water, including chitosan [18], montmorillonite [19], activated carbon (AC) [20], and nanotubes of carbon [21]. However, the wide use of these materials is restricted by their high fabrication costs, dumping, and regenerating costs [22,23]. Therefore, several attempts have been made to develop cost-effective and eco-friendly adsorbents. For example, many studies focused on coating sand particles with different waste materials, such as cement kiln dust, and use these particles as a promising adsorbent for the removal of organics and heavy metals from water and wastewater [7,24].

In this context, the current study aims to develop a new adsorbent made from sand coated with calcium ferric oxides; the latter was fabricated using wastepaper sludge ash (WPSA). It is noteworthy that the WPSA was used in this study as a source of calcium for many reasons; firstly, its chemical composition of the WPSA is rich in calcium [25]. Secondly, huge amounts of WPSA are produced annually worldwide [26]. Finally, local authorities in the UK and Europe encourage recycling paper waste [27,28]. Paper mill sludge is the semi-solid slurry gathered in effluent treatment units during deinking and repulping.

## 2. WPSA at a Glance

The sludge of paper mills is burned in combined heat and power (CHP) reactors to reduce the volume of the paper sludge (80–90% savings) and also to recover some manufacturing energy by co-combustion with biomass [29]. Combustion of the sludge produces a significant amount of ash that is categorised as trash in the UK and other countries. For instance, it was reported that forty paper mills in the UK produce 140,000 tonnes of WPSA yearly [27]. Due to the lack of recycling or recovery options, paper sludge and WPSA are mostly disposed of in landfills, which is a costly option. For example, several EU countries charge 15-€70 per tonne of non-hazardous solid waste storage, and the UK applies a tax of 3-£96 per tonne. Therefore, the WPSA needs efficient management systems, such as eco-friendly recycling options. Although there are many trials to recycle paper waste in

the UK and worldwide, the local markets are unable to recycle all the produced paper waste due to its significant quantities. Due to its high CaO concentration, WPSA has been identified as a promising source of calcium ions [25,26]. Therefore, this study has utilised WPSA as a cost-effective and eco-friendly source of calcium.

## 3. Previous Uses of Calcium Ferric Oxides

The literature shows many trials to synthesise different types of calcium ferric oxides. For instance, Chatterjee and Chakraborty [30] synthesised $Ca_4Fe_9O_{17}$ using a mixture of potassium ferrocyanide trihydrate and calcium acetate hydrate. The dry mixture was thermally decomposed at a temperature of 700 °C and then used as a photocatalyst for the degradation of organic dyes. Sadrolhosseini et al. [31] prepared $CaFe_2O_4$ using $Fe(NO_3)_3 \cdot H_2O$, $Ca(NO_3)_2 \cdot H_2O$ and polyvinyl alcohol by the thermal treatment method (at temperatures of 362 and 860 K). The produced calcium ferric oxides have been used to remove the Methylene blue and Methylene Orange dyes from water. Vanags et al. [32] made another trial to synthesise $Ca_2Fe_2O_5$ using $Fe(NO_3) \cdot 9H_2O$ and $Ca(NO_3)_2 \cdot 4H_2O$ by the sol-gel auto-combustion method at temperatures ranging between 300 and 800 °C. The produced adsorbent exhibited high efficiency in the degradation of methylene blue from water. Additionally, some studies focused on the development of nano-particles of the $CaFe_2O_4$; for example, Sulaimana [33] synthesised $CaFe_2O_4$ nano-particles by mixing a 1:1 molar ratio of nitrate $Ca(NO_3)_2$ and ferric nitrate $Fe(NO_3)_3$, then calcinated at a temperature of 550 °C. However, these nano-particles were applied in drug delivery.

The above short literature review shows the application of calcium ferric oxide as a coating layer for inert materials, such as sand, though its use as an adsorbent for water treatment has not yet been investigated. Additionally, the novel difference, to the knowledge of the authors, between the current study and the literature is that the previous studies used elevated temperatures up to 800 °C to fabricate the calcium ferric oxide, which is not an eco-friendly method, while the present study used a temperature of 95 °C (for drying purposes). Finally, the present study utilised a by-product (wastepaper sludge ash (WPSA)) as a source of calcium, which could make the new adsorbent an eco-friendly alternative to the traditional adsorbents. These differences between the current and previous studies could be enough of a reason to make the new adsorbent an eco-friendly alternative to traditional adsorbents. At the same time, there is still a need for further studies on the ability of this new adsorbent to remove other pollutants and investigate the effects of competitive ions on its efficiency in the removal of the targeted pollutants.

In conclusion, the authors believe this work represents the initial step for this new approach. Still, there are many steps that other researchers should take to make the new adsorbent applicable for industrial purposes.

## 4. Materials and Methods

### 4.1. Materials and Characterisation

The required amount of WPSA, Figure 1, was supplied by SAICA PAPER UK Ltd., while the sine sand sample was provided by Henry Cotton laboratory at Liverpool John Moores University, UK.

The supplied sample of WPSA was subjected to a series of initial characterisation tests, including the specific gravity, $D_{50}$, and particle size distribution. The measured values of the specific gravity and the $D_{50}$ of the WPSA sample were 2.5 and 13.34 μm, as measured by a Quantachrome helium pycnometer and a Laser Diffraction Particle Size Analyser (LS 12230), respectively. The particle size distribution of the WPSA sample was measured through the sieve analysis, and the results are shown in Figure 2.

The fine sand sample was sieved and then washed to ensure cleanness and extraction of the unwanted sizes of sand particles before using them in the coating process. The mean diameter, porosity, specific gravity, and hydraulic conductivity of the sieved sample of sand were 1013 μm, 0.37, 2.685, and $4.719 \times 10^{-1}$ cm/s, respectively.

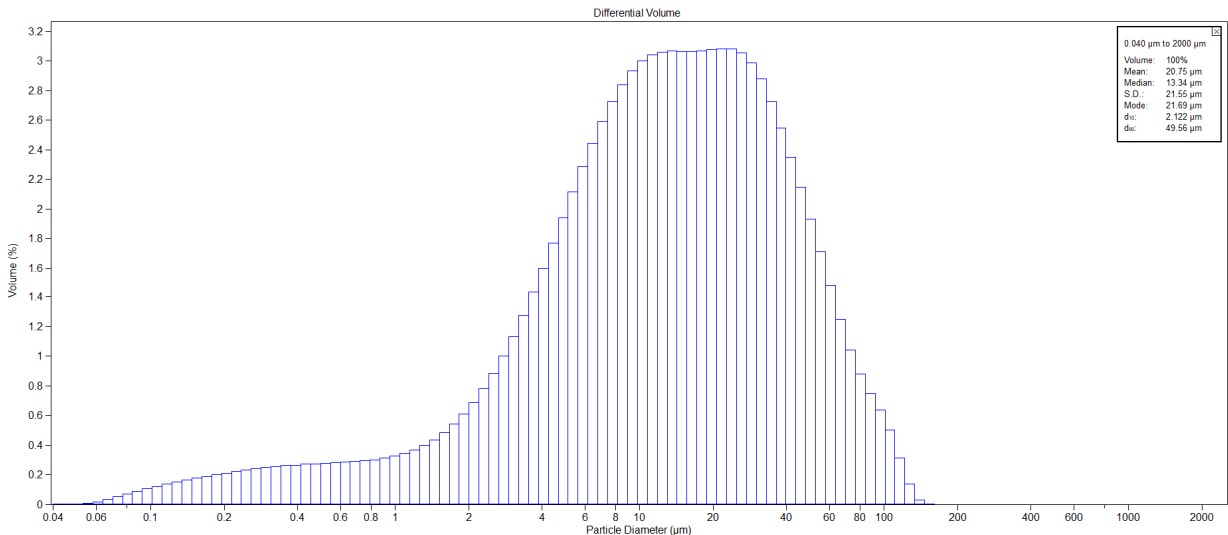

**Figure 1.** Wastepaper sludge ash.

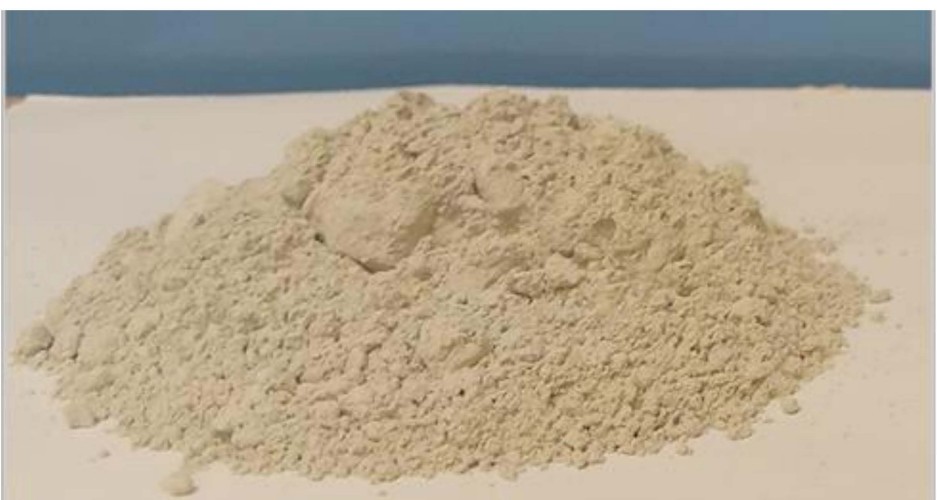

**Figure 2.** Particle size distribution for wastepaper sludge ash (WPSA).

The rest of the physical properties of both WPSA and sand samples were measured, and the results are shown in Table 1. The Chemical properties of the WPSA were determined using the XRF test, and the results are shown in Table 2.

**Table 1.** Physical characteristics of WPSA and sand.

| Property | WPSA | Sand |
|---|---|---|
| Mean diameter (µm) | 13.34 | 1013 |
| pH | 12.31 | 6.76 |
| Specific gravity | 2.500 | 2.685 |
| Bulk Density (Kg/m$^3$) | 561 | 1685 |
| Moisture content (%) | 6.74 | 0.06 |
| Dry Density (Kg/m$^3$) | 525.6 | 1684.1 |
| Void Ratio | 4.08 | 0.60 |
| Porosity | 0.80 | 0.37 |

**Table 2.** XRF analysis for Wastepaper Sludge Ash (WPSA).

| Chemical Consentient | Empirical Formula | WPSA |
|---|---|---|
| Calcium oxide (lime) | CaO | 34.004 |
| Chlorine | Cl | 8.775 |
| Sulphate | $SO_3$ | 3.144 |
| Silicon dioxide (silica) | $SiO_2$ | 3.111 |
| Aluminium trioxide | $Al_2O_3$ | 3.071 |
| Sodium oxide | $Na_2O$ | 2.872 |
| Phosphorus pentoxide | $P_2O_5$ | 1.572 |
| Titanium dioxide | $TiO_2$ | 0.804 |
| Potassium oxide | $K_2O$ | 0.608 |
| Magnesium oxide | MgO | 0.502 |
| Iron (III) oxide or ferric oxide | $Fe_2O_3$ | 0.473 |
| Zinc Oxid | ZnO | 0.158 |

### 4.2. Preparation of TC Solution

The TC solution was prepared by mixing a specific weight of TC ($C_{22}H_{24}N_2O_8$) with a specific volume of deionised water to obtain the required concentration of TC (between 10 and 200 mg/L). The pH of the solution was adjusted to the desired value using a proper volume of HCl (32%) or NaOH. All chemicals used here were supplied by Merck Life Science UK Ltd., UK and used as provided without further treatment. The molecular structure of TC is illustrated in Figure 3.

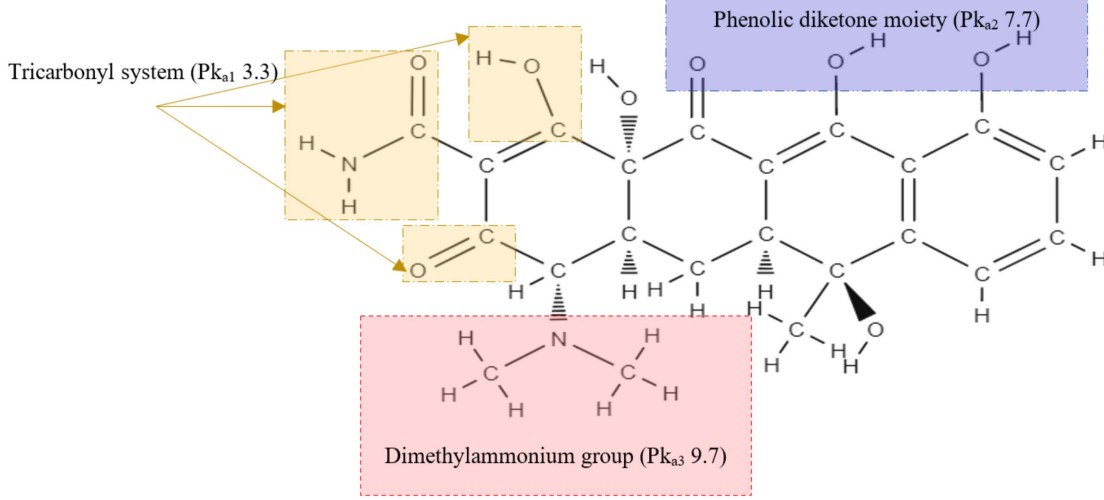

**Figure 3.** Planer view for the molecular structure of tetracycline.

### 4.3. Preparation of the Adsorbent

The new adsorbent was prepared by extracting calcium from WPSA, which was achieved by mixing 2 g of WPSA with 100 mL of water and 3 mL of 32% HCl. The mixture was continuously stirred for 3 hrs at a speed of 200 rpm. Then, the solution was filtered on Whatman No.1 filter paper, and the filtrate (rich in calcium) was mixed with 3.66 g of $FeCl_3$, 6 mL of Ethylene glycol and 3.66 g of sand. The mixture was shaken for 3 h at a speed of 200 rpm. Then, the solids were separated from the solution and left in a ventilated oven (at 95 °C) for 12 h to remove the residual moisture. The final product of this process is the new adsorbent (the coated sand particles). Figure 4, below, shows the sequence of the preparation process. It is noteworthy that the chemicals used in this stage of the work were supplied by Merck Life Science UK Ltd., Gillingham, United Kingdom and used as provided without further treatment.

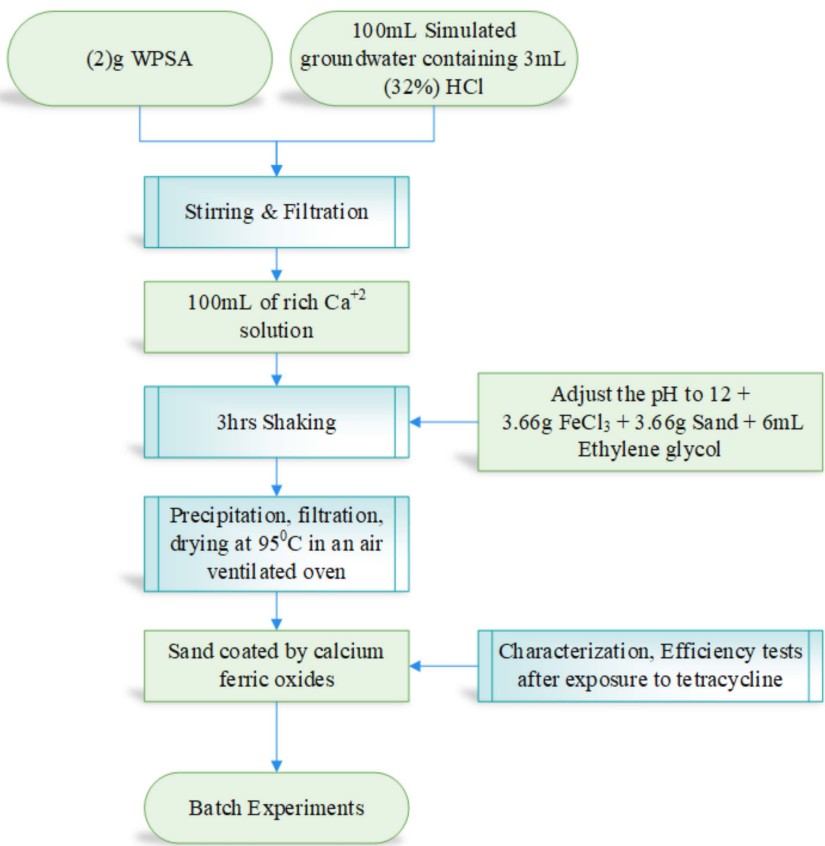

**Figure 4.** Block diagram for the synthesis of calcium ferric oxide.

### 5. Mechanisms of Adsorption Kinetics

Since adsorption kinetic models describe the sorption process onto the sorbents, they were used here to characterise the aqueous uptake rate and the required adsorption time. The adsorption occurs as a result of the physical or chemical binding between the contaminant molecules and adsorbent particles. Generally, the adsorption process and the predominant phase of the adsorption are characterised using one of three following kinetic models [34]:

#### 5.1. Pseudo-First Order

The Pseudo-First-Order model is represented by the following equation [35]:

$$q_t = q_e(1 - e^{-k_1 t}) \tag{1}$$

where ($q_e$) is the amount of contaminant adsorbed by the reactive media in the equilibrium conditions (mg/g), ($q_t$) indicates the quantity of the pollutant that has been adsorbed at a particular time ($t$) (mg/g), and $k_1$ is the Pseudo-First-Order rate of adsorption (min$^{-1}$).

#### 5.2. Pseudo-Second Order

The Pseudo-Second Order could be quantified using the following equation [36]:

$$q_t = \frac{t}{\left(\frac{1}{k_2 q_e^2} + \frac{t}{q_t}\right)} \tag{2}$$

where, $k_2$ is the rate constant of the second-order sorption (mg/(mg min)).

The kinetic model of Pseudo-Second-Order adsorption could be applied to calculate the initial sorption rate at extremely low initial concentrations.

*5.3. Intra-Particle Diffusion Model*

The adsorption process can be explained by three steps: (1) mass solution transport, in which adsorbate molecules diffuse from solution to the solution boundary layer encompassing the adsorbate; (2) film diffusion, in which adsorbate diffuses through the fluid layer encompassing the adsorbate particles; and (3) pore diffusion and adsorption. The reaction kinetics of intra-particle diffusion explain these three steps, as shown in the following equation [37]:

$$q_t = k_d t^{0.5} + Con. \tag{3}$$

where the variable ($q_t$) in (mg/g) denotes the adsorbate loading on the solid phase at any given time ($t$), $k_d$ is the intra-particle diffusion constant (mg.g$^{-1}$ min$^{1/2}$) and the *Con.* Represents the model's boundary layer constant (mg/g).

## 6. Characterisation

Several characterisation analyses were performed to investigate the properties of sorbents. A Shimadzu XRD-6000 X-ray diffractometer, made by Shimadzu Corporation (Kyoto, Japan) was used to conduct X-ray diffraction (XRD) investigations on both uncoated and calcium ferric oxide-coated sand to determine the crystal structure. The Scanning electron microscopy (SEM) tests were carried out using scanning electron microscopy, FEI Inspect-S (SEM) variable vacuum (0.1–30KV range). Particle size distribution has been performed using Beckman Coulter (LS 3 320), and the total carbon has been performed using Primacs SERIES Carbon/Nitrogen Analysers.

The content of the quantitative oxide was measured using X-ray fluorescence analysis (XRF) type Shimadzu EDX 720 analyser, made by Shimadzu Corporation (Kyoto, Japan). Furthermore, the specific gravity has been measured using a Quantachrome helium pycnometer. Finally, the concentrations of calcium and tetracycline were measured using a Benchtop Visible Spectrophotometer UV-Spectrophotometer (HACH DR-3900) made by Hach Lange GmbH Headquarter, D-40549 Düsseldorf, Germany.

## 7. Results and Discussion

*7.1. Extraction of Calcium Ions ($Ca^{2+}$) from WPSA*

As mentioned above, the extraction of the $Ca^{2+}$ ions from the WPSA was done by mixing 1.0 g of WPSA with 100 mL of deionised water containing 0.5, 1, 1.5, 2, and 2.5 mL of 32% HCl; then, the mixture was stirred at 200 rpm for 3 h. The mixture was then filtered using Whatman No.1 filter paper manufactured by Merck Life Science UK Ltd., Gillingham, United Kingdom.

The concentration of $Ca^{2+}$ ion in the filtrate was measured using a Hach Lang Spectrophotometer (DR3900) and LCK 327 cuvettes. The results showed the WPSA could produce 12,060 mg/L of $Ca^{2+}$ ion, as shown in Figure 5.

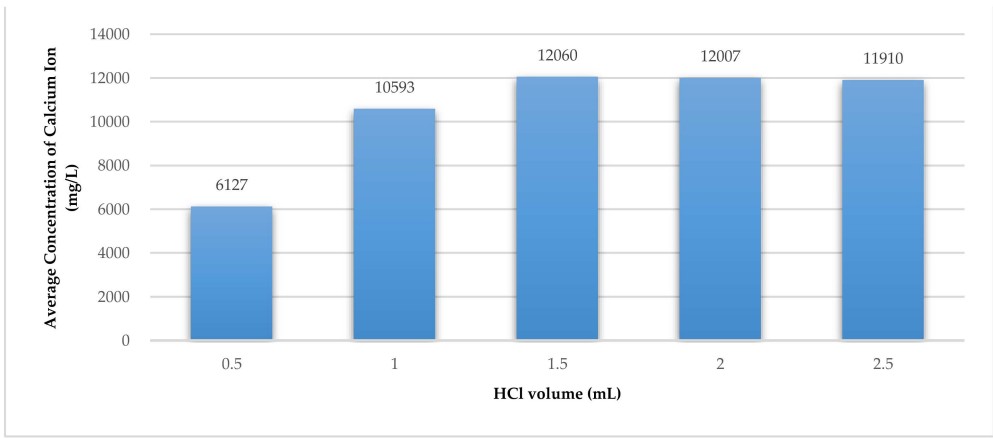

**Figure 5.** Calcium ions extracted from WPSA.

*7.2. Synthesise of the Novel Coated Sand Adsorbent (QS/CFO)*

The co-precipitation method stated by Sulaiman et al. [38] has been adopted to synthesise the new adsorbent. The performance of prepared coated sand by calcium ferric oxides has been evaluated by measuring the removal efficiency of TC by applying 0.5 g of sorbent to 50 mL of contaminated water with an initial TC concentration ($C_0$) of 50 mg/L and initial pH of 7. The mixture was shaken for 3 h at a speed of 200 rpm.

In this stage of work, the effects of six key synthesising parameters on the performance of the new adsorbent were investigated in batch experiments. The studied parameters were the Fe/Ca molar ratio, $FeCl_3$/Sand ratio, pH of the synthesising aqueous environment, Ethylene glycol dose, drying temperatures, and recoating cycles.

Initially, a set of experiments were carried out to determine the best Fe/Ca molar ratio that provides the best adsorption of TC. The optioned results showed a significant effect of the Fe/Ca molar ratio on the removal efficiency, see Figure 6a. A Fe/Ca molar ratio of 1/0.75 was selected as the best ratio in this work, as it provides the maximum removal efficiency (45.2%). Additionally, it was noticed that the use of uncoated sand achieved the lowest removal (1.9%). Another set of batch experiments was carried out under the same operational conditions mentioned previously to specify the $FeCl_3$/Sand ratio. The results of these experiments, Figure 6b, indicated that a $FeCl_3$/Sand ratio of 1/1 provided the best removal efficiency (50.5%). Therefore, this ratio was adopted in the synthesis of the new adsorbent. The effect of the pH of the synthesising aqueous environment on the removal efficiency was also investigated in batch experiments. The best removal (61.2%) was noticed at a pH of 12, as illustrated in Figure 6c. The effects of four different doses of Ethylene glycol (0 to 9 mL) on the removal efficiency of TC were studied, and it was noticed that the best removal (61.2%) was achieved at the Ethylene glycol dose of 6 mL, as illustrated in Figure 6d. In terms of drying temperatures, the coated sand was dried at four different temperatures (75, 85, 95, and 105 °C), and then the coated sand was used to remove TC. The results of Figure 6e show the best adsorption of TC happened at a temperature of 95 °C. Lee and Lee [39] explain this behaviour by changing the specific surface area; the specific surface area of the dried coated sand at a temperature of 95 °C was to be greater than the specific surface area of that dried at a temperature of 105 °C. It is noteworthy the majority of the synthesising processes in the literature were conducted at drying temperatures between 100 and 105 °C. Therefore, the present work could be a cost-effective and eco-friendly alternative to the traditional synthesising processes as it minimises the drying temperature by 5 °C. Finally, the effects of the recoating cycles on the removal efficiency of TC were investigated by repeating the coating process three times. The removal efficiency was increased from 66% (one coating cycle) to 70% (at three recoating cycles), see Figure 6f. This increase of the removal efficiency could be explained by filling uncovered spaces on the surface of the sand with calcium ferric oxide, which improves the removal efficiency. In this study, however, one coating cycle was adopted due to the relatively low increase of the removal efficiency (only 4%). In summary, the best synthesising conditions for the new adsorbent are Fe/Ca molar ratio, $FeCl_3$/Sand ratio, pH of the synthesising aqueous environment, Ethylene glycol dose, drying temperatures, and recoating cycles of 1/0.75, 1:1, 12, 6 mL, 95 °C and one coating cycle, respectively.

*7.3. Characterisation of The New Adsorbent*

A series of characterisation tests were carried out to understand the mechanism of TC adsorption by the new adsorbent. Initially, the chemical composition of the sand before and after coating was characterised using XRD analysis, as seen in Figure 7. The XRD results revealed the presence of diffraction reflections at degree 2theta (25.28, 26.28, 32.45, 36.28, 39.42, 40.25, 42.41, 44.28, 50.09, 54.10, and 56.44) that correspond to the presence of calcium ferric oxides pattern. The confirmation of the presence of calcium ferric oxides has been achieved by the PANalytical/X'Pert HighScore Plus software, version (3.0.0) for XRD powder diffraction measurements. The X'Pert HighScore Plus programme validated the synthesis of calcium iron oxides ($CaFe_2O_4$, $Ca_4Fe_9O_{17}$, $Ca_{0.15}Fe_{2.85}O_4$, $Ca_3Fe_{15}O_{25}$,

CaFe$_4$O$_7$) on the surface of the raw sand. The XRD reflections describe the new sites formed on the sand's surface, which turned the inert sand into reactive material. As a result, the new sites will oversee tetracycline removal from aqueous solutions. Based on a comparison of the "Joint Committee on Powder Diffraction Standards (JCPDSs)" and the quartz pattern obtained by this method, it seems that the silica component is the primary ingredient that is responsible for the appearance of reflections. The presence of silica oxide in the sand structure is indicated by the peaks at 21.29, 27.65, 38.26, 38.27, 43.27, 45.28, 54.26, 60.31, 64.32, 69.28, 74.63, and 77.58. The development of new reflections on the sand profile proves the formation of a new layer covering the sand; this layer will be responsible for the attraction and hunting of TC.

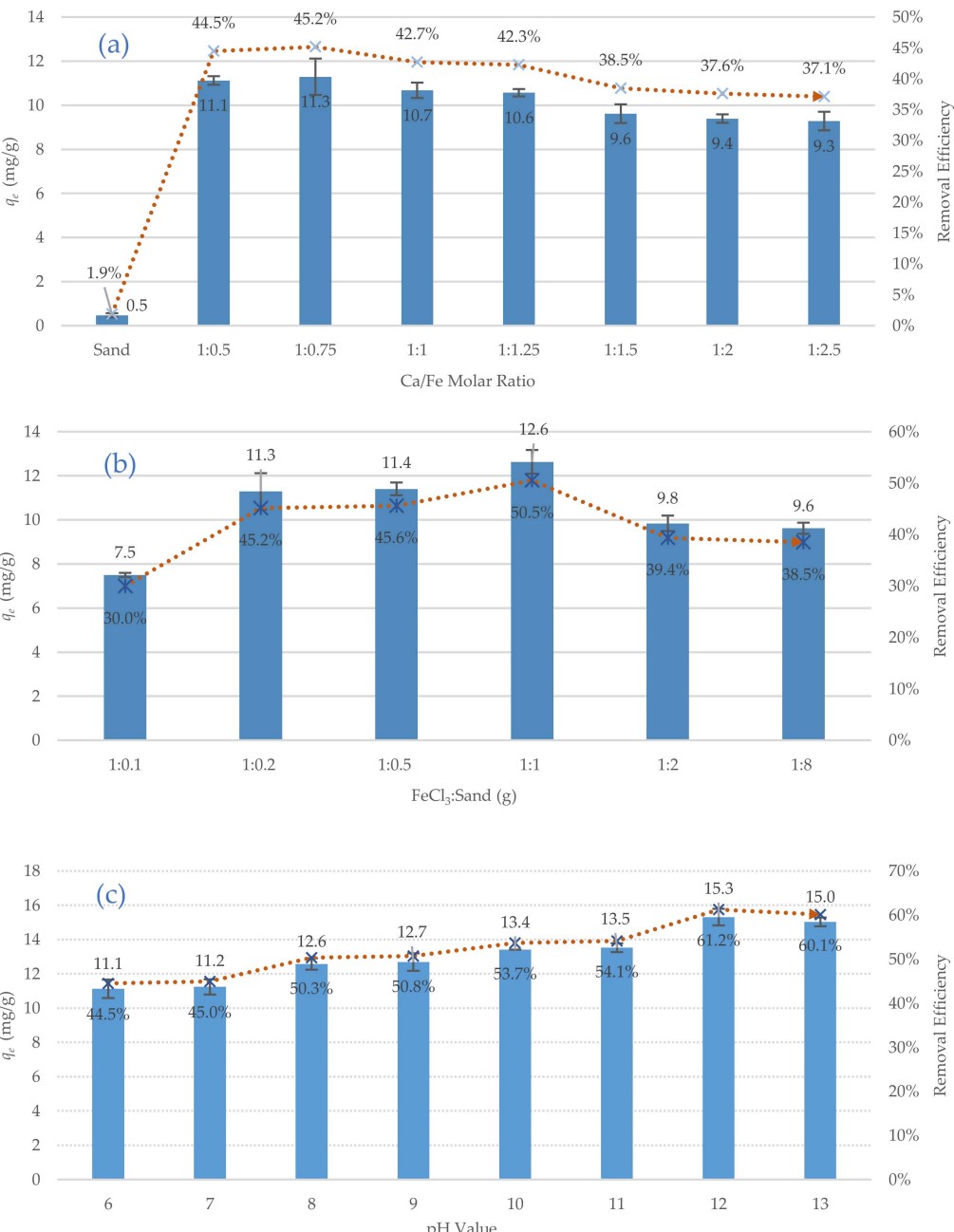

**Figure 6.** *Cont.*

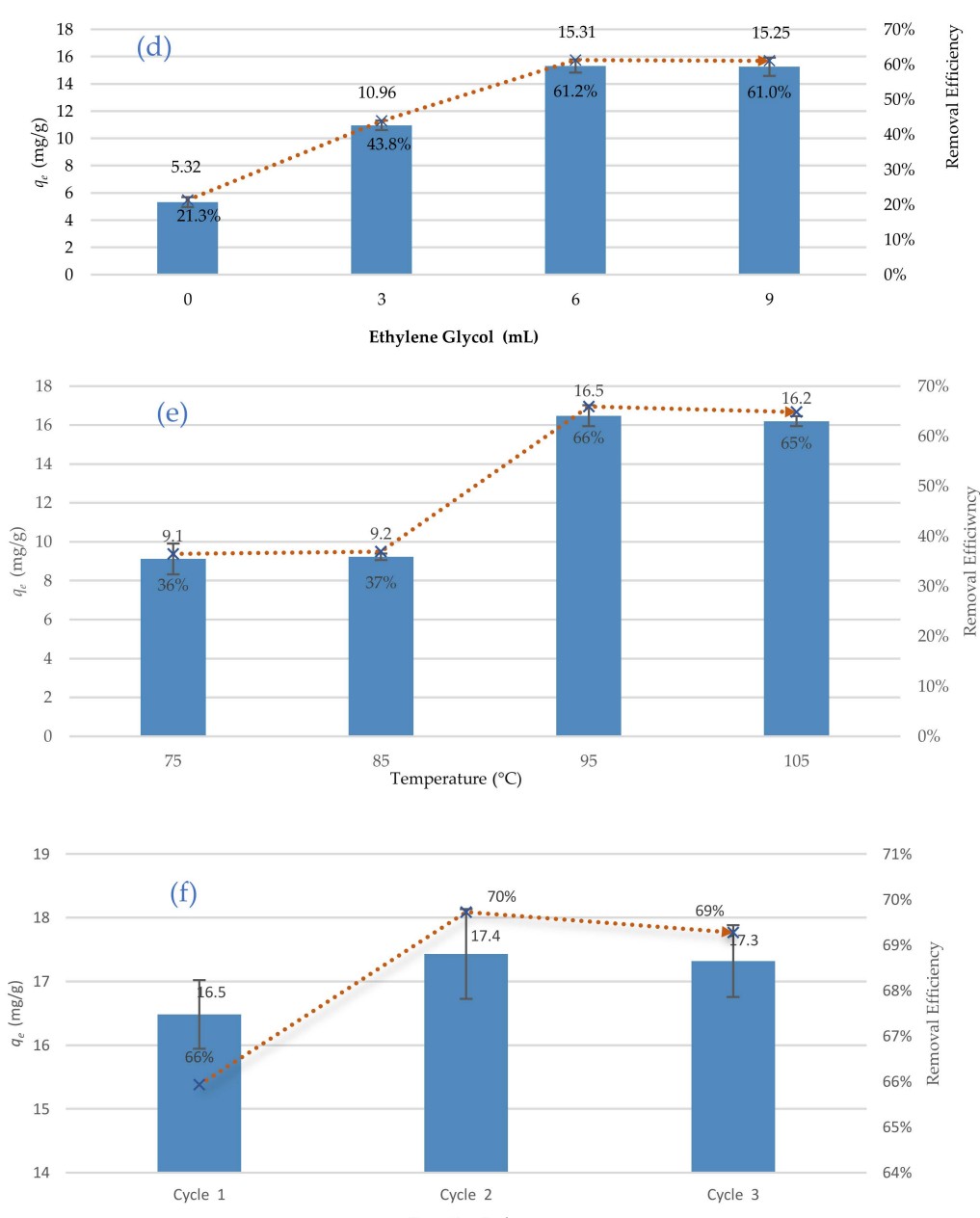

**Figure 6.** The removal efficacy, average ($q_e$) for, (**a**) different Fe/Ca molar ratio, (**b**) different silica sand percentages, (**c**) different pH values, (**d**) different Ethylene glycol values, (**e**) different drying temperatures, (**f**) multi recoating cycles.

As shown in Figure 8, SEM images illustrate the morphological properties of the new adsorbent before and after the adsorption of TC. Figure 8a shows a nonhomogeneous morphology for the porous surface of the sand, yet the surface structure is relatively compact and chaotic. Figure 8b shows the surface roughness and fractures of sand generated by the coating of the inert sand that resulted in the production of a rough layer in the coated sand consisting of coarsely aggregated surfaces (high surface-to-volume ratio) with inconsistent orientations and sizes. The latter increases the surface area of the new adsorbent, which in turn enhances the adsorption efficiency. There is also an increase of the size of cracks on the surfaces of the coated sand particles due to the influence of heating and drying temperatures.

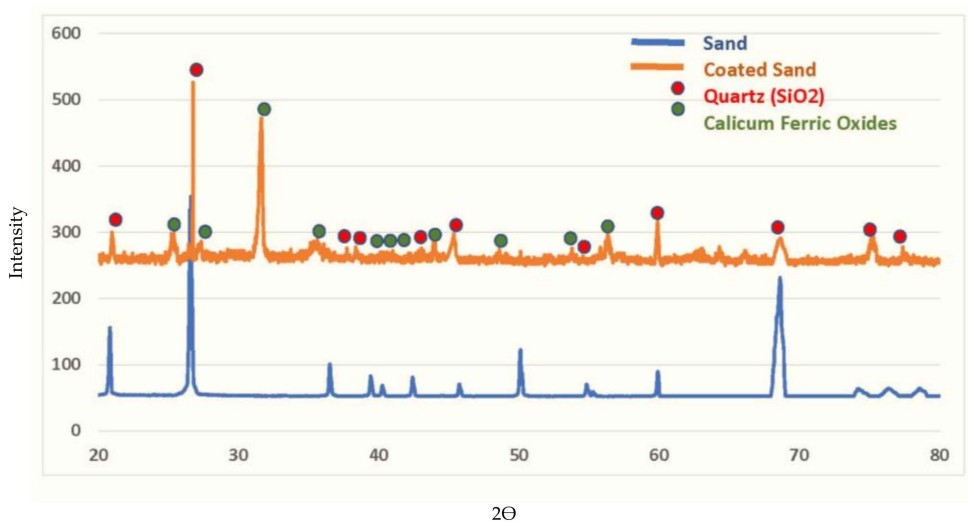

**Figure 7.** The (XRD) pattern for sand and coated sand.

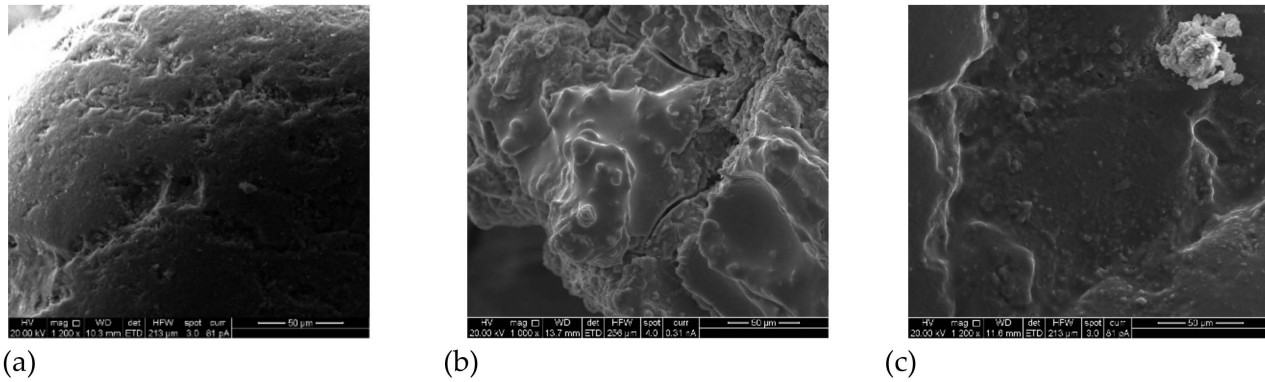

(a)                (b)                (c)

**Figure 8.** SEM images of (**a**) virgin sand, (**b**) coated sand before sorption, and (**c**) coated sand loaded with tetracycline.

Figure 8c reveals a noticeable difference in the shape of the new adsorbent before and after TC adsorption, which is related to the interaction with contaminant molecules. This finding agrees with particle size analyser data, where the mean particle size of sand increased from 1013 micrometres to 1140 micrometres before and after coating with calcium ferric oxide, respectively, as shown in Figure 9.

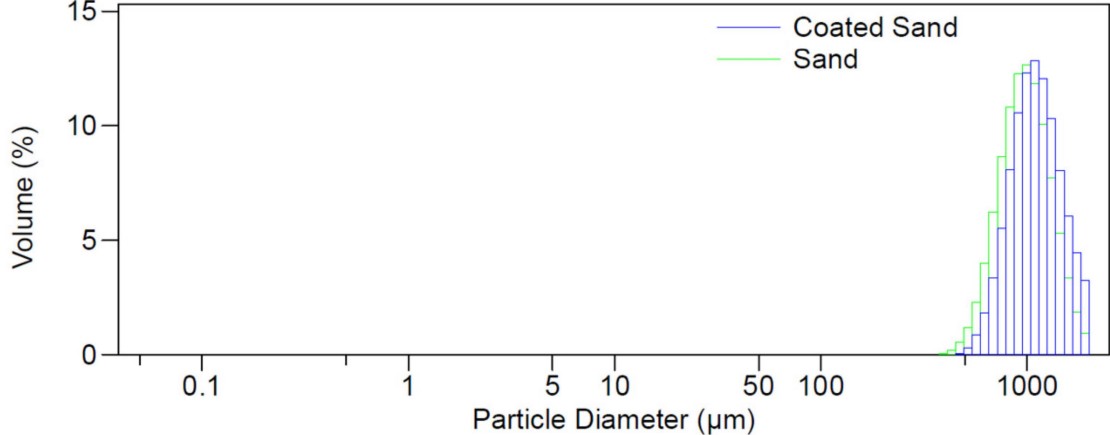

**Figure 9.** Particle size distribution for sand and coated sand.

The Primacs SERIES Carbon/Nitrogen Analyser, manufactured by Skalar Analytical B.V (Breda, The Netherlands) was used to compare the total carbon content of virgin and coated sand before and after the sorption process. This test revealed that the raw sand contains a minimal quantity of carbon (at a sensitivity of 250), as shown in Figure 10. However, the sensitivity increased from 250 to 1300 after the coating process due to the carbon added by ethylene glycol. The sensitivity then stepped up to 4300 following the TC adsorption by the coated sand. This increase of sensitivity demonstrates the efficacy of the adsorbing layer plantation and the high efficiency of TC adsorption. It is noteworthy to mention that the first three peaks in Figure 10 are blanks.

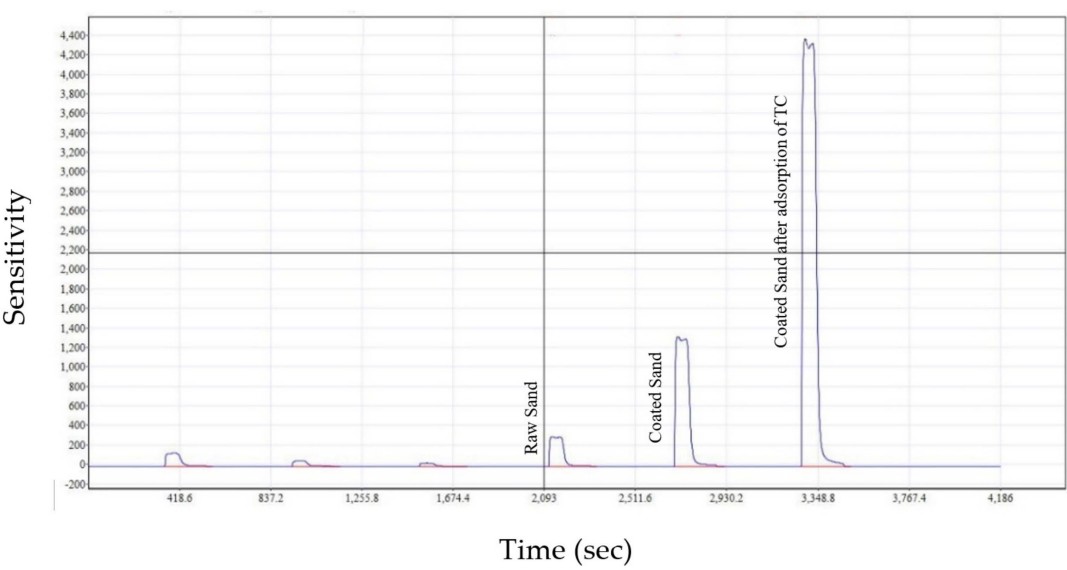

**Figure 10.** Total carbon for sand, coated sand, coated sand after adsorbing TC.

*7.4. Batch Tests*

Batch tests are used to determine the best conditions for achieving the maximum removal of TC by the developed adsorbent. These parameters included in the batch experiments were the solution's initial pH, contact time, adsorbent mass, initial concentration ($C_0$) of the TC, and agitation speed. The experiments were initiated by adding 50 mL of contaminated water ($V$) with different $C_0$ of TC (10, 50, 100, 150, and 200 mg/L) into 250 mL flasks containing 0.1 g of the new adsorbent. The flasks were shaken at a speed of 200 rpm for 3 hrs. The effects of the initial pH of the aqueous solution were examined at a range of 3 to 13, the effects of the adsorbent dose were examined at a range of 0.05 to 0.5 g/ 100 mL, and the contact time was tested up to 3 h. The solution was filtered using grade 5 Whatman filter papers to separate the solid particles. Then, a set of Hach-Lang LCK 327 cuvettes and a Hach-Lang spectrophotometer (DR3900), made by Hach Lange GmbH Headquarter (Düsseldorf, Germany) were used to determine the final concentration ($C_e$) of TC in the solution after filtration. The amount of TC loaded on the adsorbent ($q_e$, mg/g) was calculated using Equation (4), which is based on the mass balance principle.

$$q_e = (C_0 - C_e)\frac{V}{m} \tag{4}$$

where $q_e$ is the amount of adsorbate loaded on the adsorbent (mg/g), $C_0$ and $C_e$ are the initial and final concentration of TC (mg/L), respectively, $V$ is the volume of aqueous solution (L), and $m$ is the mass of adsorbent (g). The removal efficiency ($R\%$) of TC was calculated using the following Equation (5):

$$\%R = \frac{(C_0 - C_e)}{C_0} \times 100 \tag{5}$$

The results of the effects of the studied parameters are explained below.

### 7.4.1. Effects of Operating Parameters on TC Removal

*Effects of contact time*

The time required to reach equilibrium in the batch study is a critical factor in defining the diffusion of contaminants between aqueous and solid phases. Therefore, the effects of contact time on the adsorption of TC onto the new adsorbent were investigated at different contact times (0 to 250 min). The experiments were repeated for different initial concentrations of TC (10 to 200 mg/L) to understand the effects of TC concentration on the performance of the new adsorbent. The initial pH, the dose of adsorbent, and agitating speed were kept constant at 7, 0.1 g/50 mL, and 200 rpm, respectively. Figure 11a shows the results of these experiments; it can be seen from this figure that the TC elimination percentages increased quickly during the first hour of experiments, then slowed down during the rest of the experiments. Slower adsorption might be due to the increase of the number of occupied sites on the adsorbent's surface. It can be noticed that 180 min is sufficient to reach the maximum adsorption of TC; however, most of the TC was adsorbed within the first 60 min. In terms of TC concentration, it was noticed that increasing TC concentration from 10 to 200 mg/L dropped the removal efficiency from 90% to 21% respectively. The limited availability of empty sites on the surface of the adsorbent could explain this decrease of the removal efficiency with the increase of TC concentration.

Therefore, a contact time of 180 min will be used in the rest of the experiments.

*Effects of initial pH*

At room temperature, the sorption of TC onto the new adsorbent were examined at different pH levels (3 to 12) of the aqueous solution at a constant $C_0$ and sorbent dose of 50 mg/L and 0.1 g, respectively. The findings demonstrated that raising the pH value increases the removal efficacy until the pH reaches 10, and then the removal efficiency becomes stable (see Figure 11b). The ΔpH/solid addition method was utilised to evaluate the zeta potential of the new adsorbent. It was found that the point of zero charges (pHpzc) value was 5, as shown in Figure 11c, and the sorbent's surface becomes positively charged after this pH, which tends to generate an electrostatic interaction with the negatively charged pollutant. The adsorption of TC is most likely due to the π-π electron donor–acceptor interactions. The increment in TC adsorption is also related to the pore filling and the effect of H-bonding [40].

*Effects of adsorbent dose*

The effects of the adsorbent dose on the removal of TC were studied using masses of the new adsorbent (between 0.05 to 0.5 g/50 mL), keeping the values of $C_0$, initial pH, contact time, and agitation speed constant at 50 mg/L, 7, 3 h, and 200 rpm, respectively. Figure 11d shows that increasing the new adsorbent dose from 0.05 to 0.3 g/50 mL significantly increases the removal of TC from 53.4% to 91.8%, respectively. However, increasing the dose to more than 0.3 g/50 mL did not increase the removal efficiency. Therefore, a dose of 0.3 g/50 mL will be used in the rest of the experiments.

*Effects of agitation speed*

The change in TC removal with the agitation speed was studied at different speeds (0 to 250 rpm), keeping the rest of the parameters constant at the optimum values obtained from the previous experiments. The obtained results are shown in Figure 11e, which shows that the lowest removal of TC (35%) was obtained at 0 rpm (no agitation), while the maximum removal (91.8%) was obtained at a speed of 200 rpm. The increase of the TC removal with the agitation speed could be explained by the fact that increasing the agitation speed resulted in appropriate contact between the TC and the adsorbent.

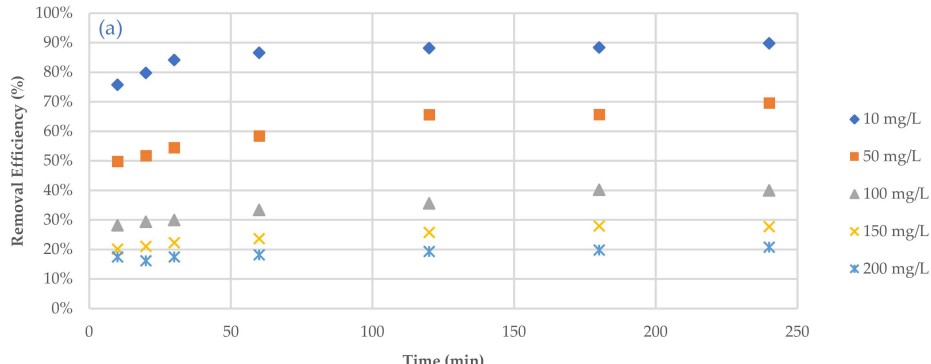

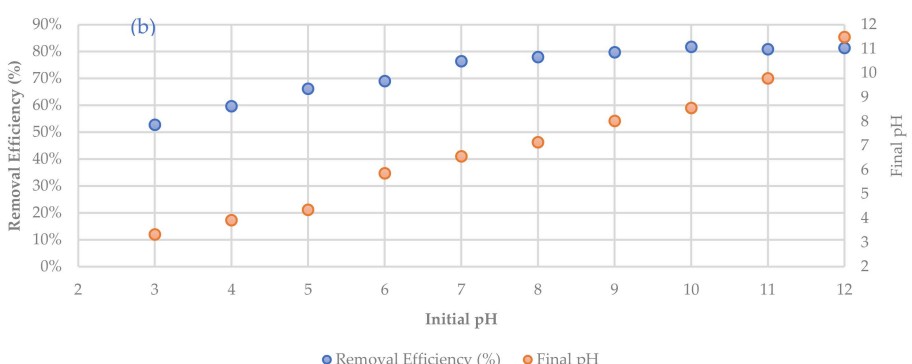

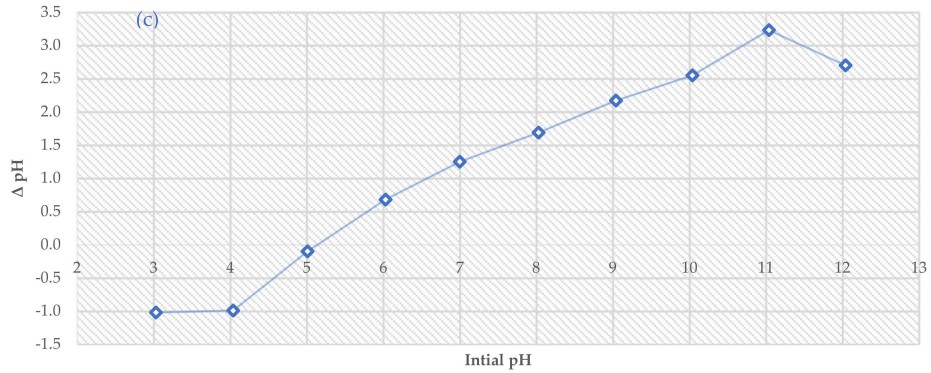

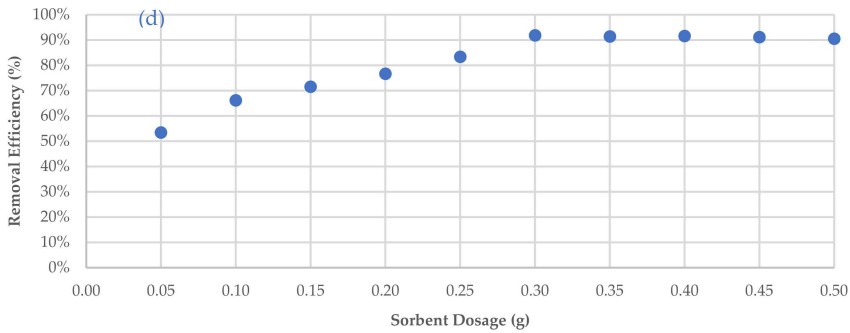

**Figure 11.** *Cont.*

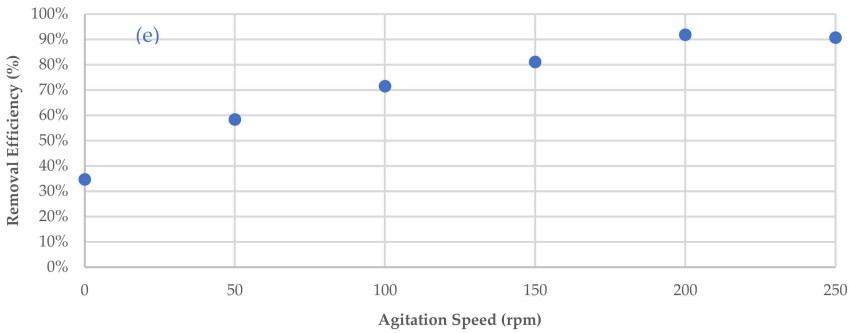

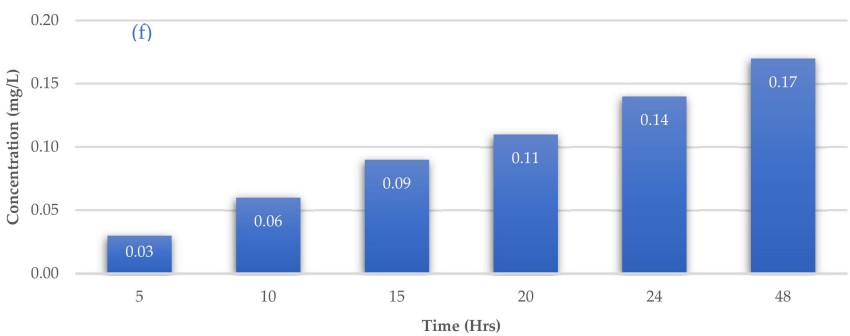

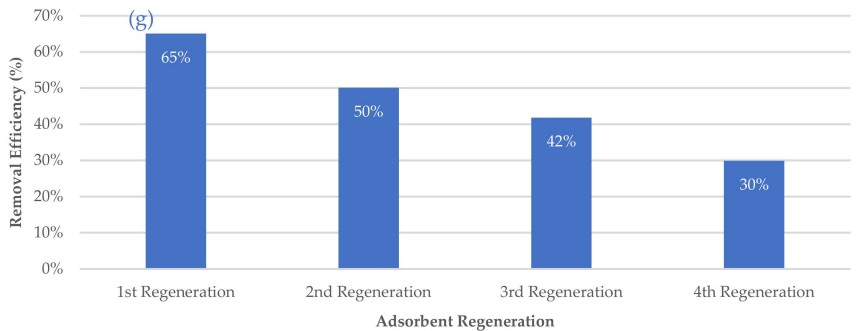

**Figure 11.** Effects of:(**a**) contact time and initial concentration, (**b**) initial pH, (**c**) point zero charge, (**d**) sorbent dosage, (**e**) agitation speed, (**f**) desorption outputs, (**g**) Adsorbent regenration.

### 7.4.2. Desorption Kinetics

Calculating the desorption properties is crucial in evaluating the adsorbent's potential and efficacy. Therefore, the desorption of the TC that adsorbed on the new adsorbent was measured by soaking 0.3 g of saturated new adsorbent (used in the removal of TC of the solution in the previous experiments) in 50 mL of deionised water for an extended contact time of 2880 min (48 h). The concentration of the TC in the deionised water was measured using a Hach-Lang spectrophotometer and LCK 327 cuvvets. The results of the desorption tests, as shown in Figure 11f, proved that the maximum released amount of TC into the deionised water after 2880 minutes did not exceed 0.17mg/L, which demonstrates that the forces between composite sorbent and TC are quite strong and that this sorbent is effective and suitable for removing TC from water.

### 7.4.3. Adsorbent Regeneration

The regeneration of the synthesised new adsorbent must be assessed to determine whether the used adsorbent might be reused again or not. Therefore, 0.5 M HCl was used to wash the used adsorbent to extract the adsorbed TC, and deionised water was then used to

wash the adsorbent. The washed adsorbent was then used to remove TC from water under the optimum operating conditions obtained from the batch experiments. This process was repeated four times (four regeneration cycles). The results of TC removal efficiency after each regeneration cycle are shown in Figure 11g, which shows that the TC removal efficiency declines after each regeneration cycle, where it declined from 65% to 30% after one and four regeneration cycles. According to the results, the produced adsorbent could be regenerated a few times before the final disposal.

### 7.5. Kinetic of Adsorption

Kinetic studies of the variation of TC adsorption with contact time onto the new adsorbent at various initial concentrations of TC were fitted with non-linear analysis using the "Solver" tool in Microsoft Excel 365. The kinetic experiments were carried out under the same optimal conditions obtained from the previous batch experiments.

Table 3 shows the constants of the kinetic models according to Equations (1)–(3) derived from the fitting process. Statistical metrics (sum of squared error (SSE) and coefficient of determination ($R^2$)) are also derived to describe the convergence between the theoretical models and the experimental data, as shown in Figure 12a,b. Kinetic fitting in this figure and Table 3 revealed that the Pseudo-Second-Order model was the best in expressing the adsorption process of TC because it has the largest $R^2$ and the smallest SSE. Furthermore, the measured adsorbed quantity of TC ($q_e$) was close to the predicted values, which is evidence of the model's applicability. These results confirmed that the TC removal is owing to chemical bonds (i.e., chemisorption). However, the kinetics models alone are insufficient for defining the processes involved in the sorption process; hence, the intra-particle diffusion model was employed in this study.

**Table 3.** Kinetics parameters for tetracycline adsorbed by sand coated with calcium ferric oxides.

| Kinetic Model | Parameter | $C_0$ (mg/L) | | | | |
|---|---|---|---|---|---|---|
| | | 10 | 50 | 100 | 150 | 200 |
| **Pseudo first order** | $k_1$ (min$^{-1}$) | 0.193 | 1.805 | 0.117 | 0.125 | 0.147 |
| | $q_e$ (mg/g) | 4.338 | 14.843 | 18.002 | 19.136 | 19.968 |
| | $R^2$ | 0.992 | 0.896 | 0.896 | 0.947 | 0.971 |
| | SSE | 0.121 | 22.351 | 22.351 | 17.030 | 10.057 |
| **Pseudo second order** | $q_{exp.}$ (mg/g) | 4.410 | 16.420 | 20.100 | 21.000 | 21.960 |
| | $k_2$ (g/mg min) | 0.113 | 0.013 | 0.010 | 0.011 | 0.014 |
| | $q_e$ (mg/g) | 4.479 | 16.728 | 19.308 | 20.391 | 21.018 |
| | $R^2$ | 0.999 | 0.981 | 0.968 | 0.981 | 0.991 |
| | SSE | 0.012 | 4.131 | 9.090 | 6.068 | 3.147 |
| **Intra-particle diffusion** | Portion (1) | | | | | |
| | $K_d$ (mg/g min$^{0.5}$) | 0.118 | 0.481 | 0.567 | 0.587 | 0.659 |
| | $Con.$ (mg/g) | 3.463 | 10.903 | 12.168 | 13.307 | 14.395 |
| | $R^2$ | 0.916 | 0.993 | 0.969 | 0.979 | 0.967 |
| | Portion (2) | | | | | |
| | $K_d$ (mg/g min$^{0.5}$) | 0.019 | 0.330 | 0.478 | 0.429 | 0.336 |
| | $Con.$ (mg/g) | 4.186 | 12.287 | 12.957 | 14.632 | 16.497 |
| | $R^2$ | 0.939 | 0.901 | 0.903 | 0.905 | 0.934 |

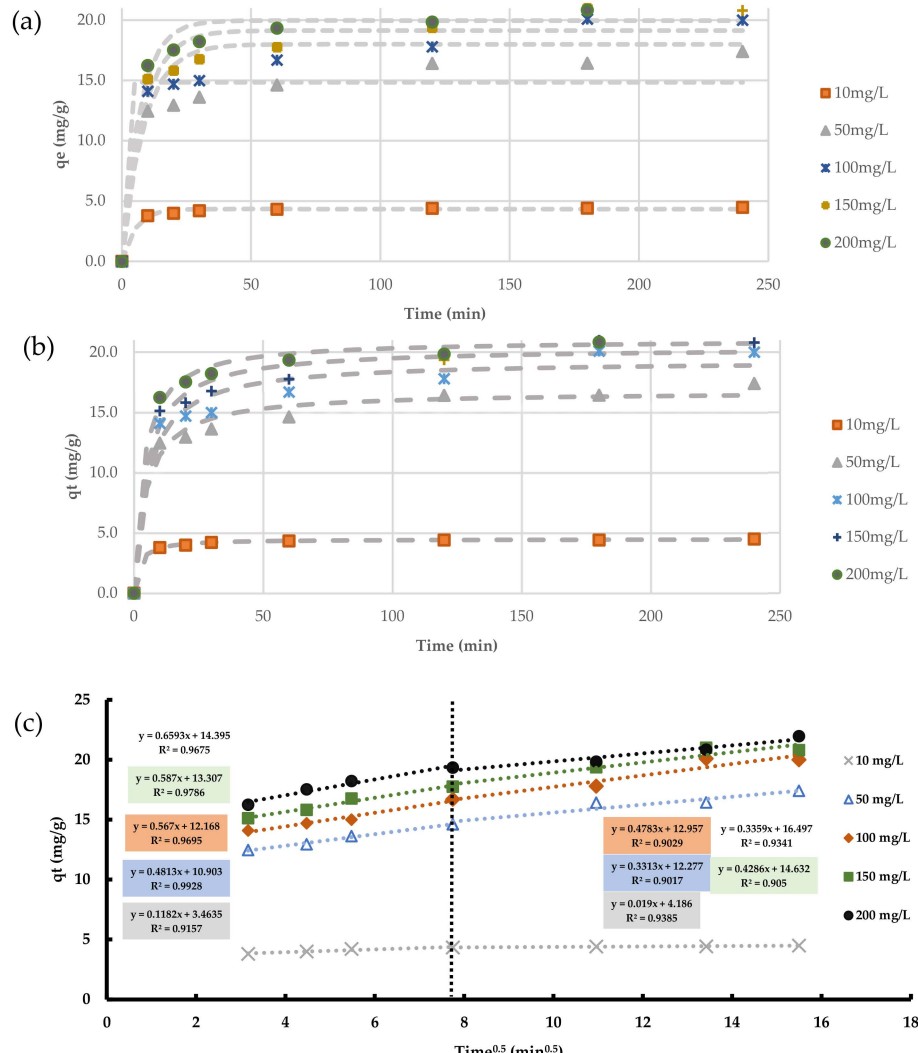

**Figure 12.** Asorption kinetics: (**a**) Pseudo first order (**b**) Pseudo second order (**c**) intra-particle diffusion model.

The intra particle diffusion model (Equation (3)) is fitted with experimental observations to specify the contribution of intra-particle diffusion and film diffusion to the contaminant adsorption. The results are presented in Figure 12c. Since the relationship between $q_t$ and $t^{0.5}$ is linear, intra-particle diffusion should occur, and this diffusion is the rate-controlling process. However, Figure 12c shows the intercept for this plot is not zero, which means another process other than intra-particle diffusion has led to the adsorption of TC.

Tetracycline (TC) is a group of broad-spectrum antibiotics that are amphoteric due to the presence of the dimethylammonium group, phenolic diketone moiety, and the tricarbonyl system. It exists primarily as a cationic ($TCH_3^+$) at pH less than 3.3, zwitterionic ($TCH_2^{\pm}$) between pH 3.3–7.7, and anionic ($TCH^+$) and ($TCH_2^-$) at pH greater than 7.7, The Zeta potential test proves that the surface charge of the (CFO-SS) is negative under (5) and positive over (5); this will enable the QS/CFO to interact with ($H^+$) and ($O^-$) ions on the TC, which leads to an electrostatic attraction between the TC and the (QS/CFO) surface. Furthermore, the SEM images proved the increment of surface roughness of the coated sand due to the deposition of calcium ferric oxides and the drying process. The latter improves the interlock of the TC molecules within the surface of the coated sand through intraparticle diffusion; this phenomenon was also proved by the intraparticle diffusion kinetics. Additionally, the iron and calcium ions on the surface of the coated sand react with

a pair of electrons and π-bonding electrons in TC functional groups (hydroxyl, carbonyl, and amino groups), forming TC-Fe, TC-Ca complexes to strengthen the cation-π bonding.

## 8. Conclusions

This article investigates the development of a sustainable and eco-friendly adsorbent via the reaction of calcium derived from wastepaper sludge ash with $FeCl_3$. Based on the results of the batch experiments, the new adsorbent showed a good ability to remove TC from water, and the best performance of the new adsorbent (achieved TC removal of 92%) can be obtained at pH 10, agitation speed of 200 rpm, and contact time of 3 hrs. Furthermore, the outputs of the kinetic study indicated that the adsorption of TC on the new adsorbent is via chemical forces, and the Pseudo-Second-Order model is suitable for expressing the adsorption of TC on the new adsorbent.

Finally, future studies are still needed to investigate the applicability of the new adsorbent for the removal of other pollutants. Additionally, it is recommended to investigate the possibility of using other by-products as sources of calcium or other useful ions to remove antibiotics from the aquatic environment.

**Author Contributions:** Conceptualisation, O.A.-H., K.H., E.L., T.M.Č., I.N. and A.A.H.F.; Methodology, O.A.-H. and K.H.; software, O.A.-H.; validation, E.L., T.M.Č., I.N.; formal analysis, O.A.-H., E.L., T.M.Č., I.N. and A.A.H.F.; investigation, O.A.-H.; writing—original draft preparation, O.A.-H., K.H., E.L., T.M.Č., I.N. and A.A.H.F.; writing—review and editing, K.H., E.L., T.M.Č., I.N. and A.A.H.F.; supervision, K.H., E.L., T.M.Č., I.N. All authors have read and agreed to the published version of the manuscript.

**Funding:** This research received no external funding.

**Institutional Review Board Statement:** Not applicable.

**Informed Consent Statement:** Not applicable.

**Data Availability Statement:** The data presented in this study are included in the article. Further inquiries can be directed to the corresponding author.

**Conflicts of Interest:** The authors declare no conflict of interest.

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
