# Peer review of "Kinetic and Equilibrium Isotherm Studies for the Removal of Tetracycline from Aqueous Solution Using Engineered Sand Modified with Calcium Ferric Oxides"

_environments, doi:10.3390/environments10010007_

Round 1

Reviewer 1 Report

Title: Kinetic and isotherm studies for the adsorption of tetracycline by an Engineered Sand Coated with Calcium Ferric Oxides (QS/CFO) nanolayer in an aqueous environment.

 Authors: Osamah Al-Hashimi, Khalid Hashim, Edward Loffill, Ismini Nakouti, Ayad A.H. Faisal, Tina Marolt Čebašek.

 The authors developed a new adsorbent to remove tetracycline from polluted water. Tetracycline is a common human antibiotic but is more frequently used for veterinary purposes. The adsorbent is based on quartz sand coated with calcium ferric oxide. Calcium was extracted from cement kiln dust waste or wastepaper sludge ash to synthesize the nanoparticles. Therefore, part of this material is obtained using byproducts materials.

 To characterize the adsorbent, the authors used several characterizations: XRD, SEM, particle size distribution, X-ray fluorescence spectroscopy (XRF), and UV-VIS. Batch experiments were done to analyze the adsorption capacity of the novel adsorbent. The batch experiments analyzed the dependence on sorbent mass, tetracycline concentration, agitation speed and pH.

 The result is that sand coated with calcium ferric oxides can remove tetracycline.

 However, the manuscript, as it is, is impossible to read and therefore, I have to reject it.

 1) The manuscript has several mistakes, which prevent a precise reading. For instance, on page 2, Figure 1 is cited, but this doesn't seem right.

2) There are also several language mistakes (see examples below in minor comments). The authors should extensively revise the logic of the phrases and the language of the entire manuscript.

3) The title should be revised. “monolayer”? “in an aqueous environment”?

4) In the introduction section, this study's background was not fully provided.  Even for other purposes, are any previous attempts to synthesize sand coated with calcium ferric oxides? If any, it should be cited in the introduction.

5) Reading the manuscript (page 3), it is unclear if WPSA, CKD, or both were used to prepare the calcium-coated sand. I think both sources were used.

6) On page 3, the authors indicate “batch and continuous column experiments”, but no column experiments are done.

7) Why is the characterization of calcium ferric oxide-coated sand in the last part of the manuscript? This should be located before the batch experiments. 

8) The paragraph on page 15, starting with “the chemical composition....” has to be rewritten. 

9) Section 5.4, the kinetics of sorption. This section has to be rewritten entirely. What did the authors learn about fitting with the PFO and PSO models?

10) Page 16, Fig 10 should be Fig. 9, Fig 10 is. Therefore, Fig, 10 is not cited.

As I mentioned above, this manuscript should be rewritten completely (not only search for language mistakes). The authors should follow a logical way to explain their results. The use of models to fit their results should carry some explanation of the results.

Examples of language mistakes. There are many examples but I only write some of them.

1)           Abstract: a queues environment?

2)           Page 2, Figure 1 is not correct.

3)           On Page 4, delete the word “manageable”

4)           On page 4, all the “were” after an equation should be “where”

5)           Page 4, “to organize a tetracycline aqueous solution...” is a phrase without sense.

6)           Page 5, re-write “the core purpose of this investigation..... environment.” 

7)           Page 5, re-write “the resulting mixing....

8)           Page 15, no sense “The development of new reflections on the sand profile that the nanoparticles plantation was successful”

9) Page 13, “Ketices fitting”?

Author Response

Authors' response to comments from reviewer 1:

Seq.

Reviewer Comments

Author response

Notes

1)        

The manuscript has several mistakes, which prevent a precise reading. For instance, on page 2, Figure 1 is cited, but this doesn't seem right.

Corrected.

2)        

There are also several language mistakes (see examples below in minor comments). The authors should extensively revise the logic of the phrases and the language of the entire manuscript.

Corrected by checking the paper by native-English speaker. Correction includes spelling, grammar, sentence structure and terminology corrections.

3)        

The title should be revised. “monolayer”? “in an aqueous environment”?

Title updated to: (Kinetic and equilibrium isotherm studies for the removal of tetracycline from aqueous solution using Engineered sand modified with Calcium Ferric Oxides)

4)        

In the introduction section, this study's background was not fully provided.  Even for other purposes, are any previous attempts to synthesize sand coated with calcium ferric oxides? If any, it should be cited in the introduction.

Previous literature on synthesized calcium ferric oxides has been added to the literature, please refer to section 3, page 3: Previous uses of calcium ferric oxides.

5)        

Reading the manuscript (page 3), it is unclear if WPSA, CKD, or both were used to prepare the calcium-coated sand. I think both sources were used.

Corrected, CKD has been removed.

6)        

On page 3, the authors indicate “batch and continuous column experiments”, but no column experiments are done.

Corrected, only batch experiments have been performed and the article corrected accordingly. Please refer to the block diagram in figure 4.

7)        

7) Why is the characterization of calcium ferric oxide-coated sand in the last part of the manuscript? This should be located before the batch experiments. 

Characterization shifted before the batch tests

8)        

8) The paragraph on page 15, starting with “the chemical composition....” has to be rewritten. 

Rewritten by native-English speaker.

9)        

9) Section 5.4, the kinetics of sorption. This section has to be rewritten entirely. What did the authors learn about fitting with the PFO and PSO models?

The section has been rewritten and renumbered to 7.5. The learned things from the kinetics of adsorption are to discover the predominant removal of tetracycline is owing to the chemical effect between the coated sand and TC, section corrected including the following:

Kinetic fitting in this figure and table 3 revealed that the pseudo-second-order model was the best in expressing the adsorption process of TC because it has the largest R2 and the smallest SSE. Furthermore, the measured adsorbed quantity of TC (qe) was close to the predicted values, which is evidence of the model’s applicability. These results confirmed that the TC removal is owing to chemical bonds (i.e., chemisorption).

10)     

10) Page 16, Fig 10 should be Fig. 9, Fig 10 is. Therefore, Fig, 10 is not cited.

Corrected

11)     

As I mentioned above, this manuscript should be rewritten completely (not only search for language mistakes). The authors should follow a logical way to explain their results. The use of models to fit their results should carry some explanation of the results.

The article has been re-checked and rewritten by the authors to include all reviewers’ comments. All red texts are corrected. You can see in the revised manuscript that most of this article has been rewritten, and corrected to meet reviewers’ comments.

12)     

Examples of language mistakes. There are many examples but I only write some of them.

 1)           Abstract: a queues environment?

2)           Page 2, Figure 1 is not correct.

3)           On Page 4, delete the word “manageable”

4)           On page 4, all the “were” after an equation should be “where”

5)           Page 4, “to organize a tetracycline aqueous solution...” is a phrase without sense.

6)           Page 5, re-write “the core purpose of this investigation..... environment.” 

7)           Page 5, re-write “the resulting mixing....

8)           Page 15, no sense “The development of new reflections on the sand profile that the nanoparticles plantation was successful”

9) Page 13, “Ketices fitting”?

All corrected.

Reviewer 2 Report

The paper by O. Al-Hashimi reports the preparation of quartz sand-based composite adsorbent modified with Ca ferrite produced from waste paper sludge ash and its investigation in tetracycline removal from aquatic media. This composite shows a promising adsorption performance including removal efficiency more than ~90 % achieved in a rather short time – 3 h. The attracting feature of this work is utilization of by-product wastes for environment remediation. However, before publication in Environment a number of important corrections should be made (see below referee’s issues). Additionally, this Manuscript is rather bad organized and should be more structured.

1. The authors utilize an original method to prepare the sand-based adsorbent?

2. Please clarify the advantages of adsorbent.

3. What is the mechanism of the Ca ferrite attachment to the sand surface.

4. Please present the schemes illustrated the adsorbent’s formation steps and its structure.

5. What interactions are involved in tetracycline adsorption on the sand-based adsorbent?

6. Conclusions: “Based on batch experiments, this material is characterised an acceptable reaction with tetracycline-contaminated water.” What means this phrase?

7. English should be improved. For instance, “queues environment”?

8. Misprints should be corrected.

Author Response

Authors' response to comments from reviewer 2:

Seq.

Reviewer Comments

Author response

1)        

 The authors utilize an original method to prepare the sand-based adsorbent?

Yes, it is to the knowledge of the authors, this synthesized adsorbent is a new, novel adsorbent. However, a brief section for the attempts to synthesize calcium ferric oxides has been added, please refer to section 3, page 3. In addition, the gap between this research and others has been added as follows:

The above short literature review shows the application of calcium ferric oxide as a coating layer for inert materials, such as sand, and the use of it as for adsorbent for water treatment is not investigated yet. Additionally, the novel difference, up to the knowledge of the authors, between the current study and the literature is the previous studies used elevated temperatures, up to 800oC, to fabricate the calcium ferric oxide, which is not an eco-friendly method, while the present study used a temperature of 95oC (for drying purposes). Finally, the present study utilised a by-product (wastepaper sludge ash (WPSA)) as a source of calcium, which could make the new adsorbent an eco-friendly alternative to the traditional adsorbents. These differences between the current and previous studies could be enough reasons to make the new adsorbent an eco-friendly alternative to traditional adsorbents. At the same time, there is still a need for further studies on the ability of this new adsorbent to remove other pollutants and investigate the effects of competitive ions on its efficiency in the removal of the targeted pollutants. In conclusion, the authors believe this work represents the initial step in this new approach. Still, there are many steps that other researchers should take to make the new adsorbent applicable for industrial purposes.

2)        

Please clarify the advantages of adsorbent.

The new adsorbent has been synthesized using a cheap by-product waste which is the WPSA, a by-product from the paper industry. This by-product exchanged the use of manufactured, expensive chemicals. In addition, the use of by-product waste will protect the environment from a hundred thousand tonnes of waste. According to this research, it could use this waste to produce a reactive material instead of throwing it into the environment. 

This explanation has been added in section 3, page 3 as explained above.

3)        

What is the mechanism of the Ca ferrite attachment to the sand surface.

Added to the article, please see lines 289 -293 (The co-precipitation method stated by Sulaiman, et al. [38] has been adopted to synthesise the new adsorbent. The performance of prepared coated sand by calcium ferric oxides has been evaluated by measuring the removal efficiency of TC by applying 0.5g of sorbent to 50 mL of contaminated water with an initial TC concentration (C0) of 50 mg/L and initial pH of 7. The mixture was shaken for 3 hrs at a speed of 200 rpm.)

4)        

Please present the schemes illustrated the adsorbent’s formation steps and its structure.

Added, please refer to figure (4), page 6

5)        

What interactions are involved in tetracycline adsorption on the sand-based adsorbent?

Added, please refer to lines 543-556, as follows:

Tetracycline (TC) is a group of broad-spectrum antibiotics that are amphoteric due to the presence of the dimethylammonium group, phenolic diketone moiety, and the tricarbonyl system. It exists primarily as a cationic () at pH less than 3.3, zwitterionic ()) between pH 3.3-7.7, and anionic () and () at pH greater than 7.7, The Zeta potential test proves that the surface charge of the (CFO-SS) is negative under (5) and positive over (5); this will enable the QS/CFO to interact with (H+) and (O-) ions on the TC which lead to an electrostatic attraction between the TC and the (QS/CFO) surface. Furthermore, the SEM images proved the increment of surface roughness of the coated sand due to the deposition of calcium ferric oxides and the drying process. The latter improves the interlock of the TC molecules within the surface of the coated sand through intraparticle diffusion; this phenomenon was also proved by the intraparticle diffusion kinetics. Additionally, the iron and calcium ions on the surface of the coated sand react with a pair of electrons and π-bonding electrons in TC functional groups (hydroxyl, carbonyl, and amino groups), forming TC-Fe, TC-Ca complexes to strengthen the cation-π bonding.

6)        

Conclusions: “Based on batch experiments, this material is characterised an acceptable reaction with tetracycline-contaminated water.” What means this phrase?

Conclusions have been rewritten and corrected entirely.

7)        

English should be improved. For instance, “queues environment”?

Corrected by checking the paper by native-English speaker. Correction includes spelling, grammar, sentence structure, and terminology corrections.

8)        

Misprints should be corrected.

All manuscript has been rewritten by native-English speaker. Correction includes spelling, grammar, sentence structure, and terminology corrections.

All texts have been corrected, rephrased, and rewritten to meet the requirements of MDPI and reviewers’ comments.  

Round 2

Reviewer 1 Report

I think the authors improved the manuscript according to the comments in the first review. 

Reviewer 2 Report

The authors have taken into account all referee's comments.

This paper may be published in the present form.